

# Using video games for volcanic hazard education and communication.

L. Mani[1], P. D. Cole[1] and I. Stewart[1]

[1] School of Geography, Earth and Environmental Sciences, Plymouth University, Drakes Circus, Plymouth, PL4 8AA

*Correspondence to:* L. Mani (lara.mani@plymouth.ac.uk)

## ABSTRACT

This paper aims to understand whether video games (or serious games) can be effective in enhancing volcanic hazard education and communication. Using the eastern Caribbean island of St. Vincent, we have developed a video game – St. Vincent's Volcano - for use in volcano education and outreach sessions, aimed at improving resident's knowledge of potential future eruptive hazards (ash fall, pyroclastic flows and lahars). Here, we discuss the process of game development including concept design, game development through to final implementation on St. Vincent. Preliminary results for game implementation (obtained through pre and post-test knowledge quizzes) for both student and adult participants suggest that a video game of this style can be effective in improving learner's knowledge. Both groups of participants demonstrated an increase in score percentage (9.3% for adults and 8.3% for students) and when plotted as learning gains (0.11 for adults and 0.09 for students). This preliminary data could provide a sound foundation for the increased integration of emerging technologies within traditional education sessions.

## 1.   INTRODUCTION

Education and communication plays a vital role in improving resilience of vulnerable populations at risk from natural disasters (Johnston et al, 1999; Ronan and Johnson, 2003; Shaw et al, 2004; Paton et al, 2008). Conventionally, such awareness-raising activities are delivered in a number of guises - typically leaflets, posters, presentations, maps, TV and radio broadcasts. Often these educational products are aimed at school-aged children, in part due to ease of access and in part reflecting current thinking that children filter information through to their parents through informal conversations (Ronan and Johnston, 2003; Carlino et al, 2008; Sharpe and Izadkhah, 2014). However, it is becoming increasingly important to evolve existing education and communication techniques to better engage with a new generation of learners. It has been argued that, with advancements in technology, individuals today learn in ways different to their predecessors (Prensky, 2001;; Annetta, 2008; Bekebrede *et al,* 2011; Sharp, 2012). A fresh generation of learners (N-Gen or Net-Gen) are accustomed to a digital age in which information, news and entertainment are obtained instantaneously and delivered directly to them on a personal device (e.g. mobile phones, tablets and laptops). This has led to a rise in innovative techniques in the classroom, such as video games, in an effort to better motivate this new generation to learn (Prenksy, 2001).

This paper will focus on how educational video games can be used as a tool for public outreach for raising awareness of volcano hazards with at-risk communities. It reflects a recent surge in the application of so-called 'serious games' - video games whose primary purpose are educational, not entertainment - for the purposes of learning and training (Michael and Chen,


2005; Zyda, 2005; Djaouti et al, 2011), ranging from medicine to military training, and spanning everything from personal health to curriculum education. The paper considers the emergence of serious games in the realm of natural hazard education, and critically examines their role for communicating volcanic hazards. Highlighting a lack of empirical evidence to demonstrate that geohazard-related serious games promote more effective learning, the paper presents the development and testing of a serious game specifically designed to test volcanic hazard awareness among school children and adults on the Caribbean island of St. Vincent. The study evaluates the effectiveness of virtual environments as a learning technique and discusses the practical issues and challenges encountered when conducting this type of research.

## 2. PREVIOUS RESEARCH

In 2015, the United Nations formalised The Sendai Framework for Disaster Risk Reduction 2015-2030, with the goal of reducing the number of deaths, injuries and impact caused globally by disasters (human and natural). To address that goal, The Sendai Framework identifies the need for participating countries to "strengthen public education and awareness in disaster risk reduction", specifically promoting the use of social media and community mobilisation campaigns and encouraging the education of all at-risk communities (UNISDR, 2015). This reflects an acceptance within the disaster risk reduction arena that education and outreach programmes can prepare hazard-prone communities (McKay, 1984; Ronan and Johnston, 2001; Paton et al, 2008; Johnston et al, 1999; Paton et al, 2000), and a corresponding shift towards making these programmes more effective. The emergence of the leading role of social media highlights the educational advancements of digital technology, notably the use of sophisticated GIS programmes to replace 2-D maps with 3D terrain models, and the popularity of novel creative media for virtual reality animations.

### 2.1 3D Maps

Maps are one of the most common tools for education in hazardous areas, whether it be topographical maps to demonstrate landscape features or hazard maps to show areas most at risk from being affected by a natural hazard. Nevertheless, the use of maps to communicate hazard to at-risk communities can encounter problems, notably with high levels of illiteracy, comprehension, language and terminology (Donovan 2010). Haynes et al. (2007) explored local residents' comprehension of existing 2D maps and newly developed 3D maps for volcanic hazard mapping on the Caribbean island of Montserrat. Two resident groups were used, with the first asked to locate themselves and landmarks on 2D maps and the second using 3D topographic maps. Both groups were also supplied with oblique aerial photographs. The study showed there was a minor improvement for the participants using the 3D maps in their ability to locate themselves however the most significant improvement was in the understanding of the relationship between hazard and topography with the aerial photographs. In a similar study for lahar hazards at Mount Hood, Oregon, USA, participants were shown four maps and asked to complete activities for terrain interpretation, estimation of lahar travel times and evacuation routes (Preppernau and Jenny 2015); the study showed that most participants preferred the use of 3D maps, being able to interpret the terrain better and choose the more appropriate evacuation routes.

A logical progression of this realisation is how we may be able to improve hazard awareness using better resolved 3-D representations of hazardous settings. Further, understanding whether we can use of virtual reality environments to fosters even more effective spatial



thinking. Computer games are increasingly offering that potential, hence the emergence of 'serious games', defined by Michael and Chen (2005) as a game in which "education is the primary goal, not entertainment, with the intention of improving a specific aspect of learning". To be effective as a learning tool, a serious game needs to be carefully designed and

underpinned with a sound and compatible learning theory. The rise of serious games has been developed as a potential way to engage with a new generation of classroom learners – N-Gen or Net-Gen as first coined by Tapscott (1998) - who are characterised by a more technological way of life (Prensky, 2001).

**2.2    Serious Games**

The potential for serious games in disaster reduction and outreach is highlighted by the fact that one of the first purpose-built computer games was created by the UN International Strategy for Disaster Reduction (ISDR) as part of the Hyogo Framework for Action. The ISDR game, 'Stop Disasters!', was designed to educate children about preparing for disaster by

building resilient communities for a number of hazards (e.g. hurricanes, earthquakes, wildfires and flooding). The game was supported by a website of teaching materials and has been used in education programmes by the Seismic Research Centre (SRC) in the Caribbean region (SRC, pers. comm. 2015). A similar approach was taken by UNESCO and the government of Japan in designing 'Sai Fah: The Flood Fighter', a bespoke built game built as a response to devastating

floods in Thailand in 2011. Like 'Stop Disasters!', the game is a simple platform aimed at children, intended to educate them about how to recognise when flooding is likely and about the actions they should take to prepare. The game is freely available for a number of devices and has been translated into several languages for use in education programmes around the world.


Bespoke designed video games such as Sai Fah: The Flood Fighter and Stop Disasters! allow for a more interactive, engaging and tailored educational experience for the players. Another advantage is the ability of games to provide instantaneous feedback to the player, which means that misconceptions and misappropriation of knowledge can be avoided at an early stage. As is

commonplace in outreach activities, the target audience for both games is children, but the widespread and popular use of video games means that adult learners could benefit too. Sai Fah is available on many mobile devices and is run primarily as a stand-alone application. This means that adults have access and means to play the game and make themselves more aware.

In summary, given that videos games are emerging as a popular tool in disaster reduction education, it would seem timely to critically appraise whether such serious gaming offer an effective means of strengthening public awareness more broadly among hazard-prone communities. This paper does that in the context of communities on the eastern Caribbean island of St Vincent, where volcanic activity threatens a population largely unfamiliar with its

most extreme eruptive potential.

**2.3    Study Location: St. Vincent and the Grenadines, Lesser Antilles**

St. Vincent is the largest island in an archipelago that forms the Eastern Caribbean nation of St.

Vincent and the Grenadines. Most of its 110,000 inhabitants (Worldbank, 2016), live in the capital Kingstown, located to the south of the island. The northern part is occupied by an active volcanic centre, La Soufriere, which has a violent eruptive history, with significant explosive


eruptions in 1718, 1812, 1902 and 1979. Contemporary accounts and photographs of the 1902 eruption by Anderson and Flett (1903) show that much of the north of the island, including its

extensive plantations, was devastated and over 1500 people were killed. The 1979 eruption of La Soufriere was of a significantly smaller scale, causing no fatalities but forcing the evacuation of 20,000 people to shelters in the south, where they remained for many months after the eruption. Since 1979, the volcano has entered a state of quiescence and shows very little signs of life with only minor fumarolic activity within the summit crater.


Although many islanders are aware of the 1979 eruption, over half (56%) of the population are under the age of 35, meaning they have no direct experience of volcanic eruptions on the island. In contrast, with St Vincent prone to many other natural hazards, - experiencing hurricanes, flooding, earthquakes and landsliding on a regular basis - the island's residents tend

to prioritise preparedness for other extreme events at the expense of volcanic threats, which are deemed to pose less of a risk to day-to-day life, a phenomena previously noted in other areas prone to multiple natural hazards (e.g. Shaw et al, 2004; Perry and Lindell, 2008). On St. Vincent, therefore, raising awareness about volcanic hazards is vital to insure the population is motivated to prepare for a potential future eruption of La Soufriere.


In the event of a volcanic eruption of La Soufriere, it is the responsibility of the National Emergency Management Organisation (NEMO) to coordinate the emergency response. NEMO are a government department whose role is disaster preparation, mitigation and management for St. Vincent and the Grenadines. NEMO have developed the National Emergency Plan for a

variety of natural hazards that threaten the island and undertake a continual programme of public education to encourage community preparedness. In addition, the Seismic Research Centre (SRC) at the University of the West Indies monitors La Soufriere, in partnership with the Soufriere Monitoring Unit (SMU), via a remote network of instruments around the island and has responsibility for providing information regarding the volcano to the government of St.

Vincent during a potential or developing volcanic crisis.

Together, the NEMO and SRC coordinate an annual programme of education and outreach designed to commemorate the anniversary of the 1979 eruption (13 April). Called Volcano Awareness Week (VAW) this week-long programme of activities aims to supplement elements of volcano education included within different levels of the national school curriculum. The

principal activities Include:
- o Visits to many primary and secondary schools across the island to give education sessions,
- o Volcano hikes accompanied by local geologists and people that monitor the volcano
o Community-based outreach sessions open to the public.
- o Review of the National Disaster Emergency Management Plan by stakeholders.

The public outreach sessions present general information about the formation of volcanoes in the Caribbean as well as specific information about hazards that may occur during a future

eruption of La Soufriere. Robertson (2005) developed a volcanic hazard map for St. Vincent, highlighting the areas that are likely to be affected using a traffic-light colour coding (where red is the most hazardous and green the least hazardous). This hazard map is widely circulated across the island and a reference copy is printed within the island's telephone directory. The hazard map is also included in a SRC leaflet 'Volcanoes of St. Vincent' which explains about the

volcano, its history, and the monitoring network. NEMO and SRC also often conduct television



and radio broadcasts during outreach campaigns to publicise their work and to promote awareness.

St Vincent's active and diverse programme of outreach activities provides an appropriate backdrop to appraise the efficacy of emerging computer-based depictions of volcanic hazards against conventional volcano hazard education efforts. The following section describes how a bespoke computer game, St. Vincent's Volcano, was designed based on information collected from on-the-ground surveys, storyboarded on the basis of La Soufriere's historical eruptions, and developed over the course of a year by a team of developers at Plymouth University to produce an interactive game. Later sections report on the experience of trialling St. Vincent's Volcano with small user groups on the island to road test the software and identify technical improvements for future iterations of the game.

## 3.    St. Vincent's Volcano

### 3.1    Designing the game

During August 2014 a series of focus groups were held in two community groups to the north of St. Vincent (Owia and Petit Bordel), in close proximity to the volcano, to establish what end users wanted from the game. In addition, an online questionnaire of requirements was emailed to key stakeholders in volcano education on St. Vincent (UWI SRC, NEMO, SMU and The Red Cross). The main features that were frequently mentioned in the focus group discussions and the completed questionnaires are summarised in Table 1.

*Table 1. Summary of the key ideas and repetitive themes from the user requirements data collection focus groups and online questionnaires.*

This information from the St. Vincent focus groups and stakeholders established the basic concept of the game. Information about the eruptive history of La Soufriere was deemed to be important as well as details for three primary volcanic hazards that are likely to occur during future activity (pyroclastic flows, lahars and ash fall). Two historic eruptions scenarios, based on the 1902 and 1979 eruptions, were integrated into the game. The reconstructive visualisations illustrate the timeline of actual events during each eruption and are accompanied by textual descriptions and oral descriptions voiceover 'radio' recordings. The three volcanic hazards were integrated into interactive eruptive scenes with visualisations of each phenomena augmented by clickable icons that reveal brief dialogue boxes that explain in simple terms the formation and behaviour of that hazard.

An important element in the design of St. Vincent's Volcano was storyboarding the game concept as a communication pathway between designers and developers. A series of storyboards were created for each game scene, providing detailed descriptions on the look-and-feel, nature of player interactivity and navigation through the game. They also included all textural and oral descriptions provided throughout the game and detailed descriptions of how the hazardous phenomena (ash fall, pyroclastic flows and lahars) behave to enable realistic-looking visualisations to be developed.

To constitute a 'serious game' it is important that it has a robust pedagogic underpinning. In the case of the St. Vincent's Volcano game the primary pedagogical method used is Kolb's model


for experiential learning; this model involves a learner transferring an experience they have
undergone into concrete knowledge, which they are then able to apply to future learning
experiences (Kolb, 1984; Vince, 1998; Bellotti et al, 2013; Konak et al, 2014). Four stages make up
the learning cycle: concrete experience, reflective observation, abstract conceptualisation and
active experimentation. In the context of St. Vincent's Volcano, this experiential learning
component is achieved by the player reflecting on the experience and about their personal
performance through scoring systems and instantaneous feedback for completed tasks.

A secondary learning theory embedded in the game development is Swellers' 'Cognitive Load
Theory', which is based on the hypothesis that for effective learning a person's short-term
memory can contain simultaneously only a certain number of elements (Chandler and Sweller,
1991; Sweller, 1994; Bellotti et al , 2013). Ensuring that the participant's attention was not unduly
overloaded was achieved in the game by eliminating repetitive information to reduce
redundancy and engaging both auditory and visuals senses to increase the working memory
capacity.

The game development was an iterative process fed by continual feedback and testing between
designers and developers, throughout the one-year development process. The finalised version
of the game varied little from the storyboarded design, with minor differences due to technical
constraints of the software used (Unity 3D).

**3.2    Game Overview**

The completed St. Vincent's Volcano game (Fig 1.) incorporates user requirements, learning
theory and established volcanic hazard communication information (e.g. volcanic hazard map).
Three geographic perspectives are adopted, reflecting the three most populated centres in
proximity to the volcano: Chateaubelair, Georgetown and Fancy. In each of these perspectives,
there are four main scenes:

A.  Interactive island model – the player can manipulate a top-down view model of the
island to identify where they live in proximity to the volcano. The model is a realistic
visualisation built using digital elevation data and satellite imagery. The players can add
the volcanic hazard map with detailed definitions to the model to determine which
hazard zone they live in. This scene is designed to be highly interactive with the player
able to manipulate the model to explore the island.

B.  Historical eruption visualisations – La Soufriere is brought to life through a series of
visualisations depicting two historic eruptions (1902 and 1979). The visualisations are
based on the historical record and peoples personal accounts of the eruptions. The
visualisations are accompanied by textural descriptions of the activity and oral
descriptions provided as 'radio broadcasts'.

C.  Hazard training – three scenes of visualisations depicting the potential future hazards
phenomena (ash fall, pyroclastic flows and lahars). The player is guided through via five
clickable icons in each scene, which reveal snippets of information about the formation
and behaviour of each phenomenon.

D.  Volcano quiz – a multiple-choice quiz where all questions relate to information given
throughout the game. All answers are recorded through in-built game analytics.



*Figure 1. Screenshots from the St. Vincent's Volcano game. (a)The interactive island model where players can see where they live in proximity to La Soufriere and also view the volcanic hazard map. (b) A scene from the 1902 historical eruption visualisation including textural descriptions of the events as they unfold. (c) Hazard training section of the game for pyroclastic flows and surges*
*with textural descriptions revealed when icons are clicked. (d) An example of a question provided during the volcano quiz at the end of the game.*

## 4.    Field Implementation with Target Audiences

To ensure that the volcano game was appropriate for the target audience, St Vincent's Volcano
was trialled on the island over a 6 week period (March to May 2015) that coincided with Volcano Awareness Week (VAW). Two types of outreach session were organised - student learners and adult learners.

### 4.1    Student Learners
The Ministry of Education for St. Vincent and NEMO recruited schools through a circular email prior to the VAW activities explaining about the study using the game. For the 2015 VAW activities, schools primarily in the 'Green' volcanic hazard zone (based on the Robertson 2005) established volcanic hazard) were targeted for outreach - primarily schools located in the Kingstown area of the island and surrounding towns. Since many students travel to the capital,
Kingstown, for school but often do not reside there, all students participating in the study were asked to provide their location of residency on St. Vincent (Fig. 2). In total, 13 secondary schools were involved in the VAW activities with data being collected from 6 of these schools (due to time and facility constraints). Sessions typically comprised between 20-35, Grade 4 (14-15 years old) students, which was deemed optimal for the study.

*Figure 2. Map showing the location of residency for all participants used in this study.*

Each session was run dependent on the time slot and facilities available and the number of students per session. The students were either asked to listen to a presentation by a member of
the UWI SRC outreach team on volcanic hazards and St. Vincent, or play the St. Vincent's Volcano game, or both. Students were able to ask questions throughout the session from any member of the outreach team and support was given when required.

The presentation some of the students received included basic information on the tectonic
regime in the Caribbean and its link to the formation of volcanoes, the historical eruptions of La Soufriere and descriptions of the types of eruptive phenomena that could be experienced on St. Vincent in the future (e.g. pyroclastic flows, lahars and ash fall). Further information included how the volcano is monitored and about the organisations who are responsible for this (UWI SRC and SMU).

### 4.2    Adult Learners
With the assistance of community leaders, four adult sessions were organised across the island in the towns of Georgetown, Chateaubelair, Fancy and Kingstown with a total of 25 participants. The participants were recruited through word of mouth and through the community leaders
inviting community members they thought would be interested in being involved. Adult sessions were held in a variety of location including a pre-school, secondary school, community centre and civil offices (Fig. 3).


*Figure 3. Photographs taken during adult game testing sessions (a) at a pre-school in Fancy and*
*(b) at the community centre in Georgetown.*

During the adult sessions, the participants were asked only to play the game with no supporting presentation. Participants were able to ask questions throughout the session and assistance was given to those that found difficulty with using the game. After each session, the participants
were fully debriefed and able to ask any further questions they had relating to the information they had read.

### 4.3 Collecting data on player response

The implementation sessions provided a valuable opportunity to collect data about the
responses of those playing the volcano game that could evaluate its effectiveness as a learning or awareness-raising tool. Qualitative and quantitative data about player response were obtained through three contrasting methods.

The main source of data collection was through the completion of knowledge quizzes before
and after all of the sessions took place. The quizzes contained questions relating to volcanoes and St. Vincent including definitions for different eruptive phenomena and asking to name the volcano and historical eruptions of St. Vincent. The pre and post-test quizzes comprise the same questions although were slightly re-worded and the order of questions was altered to stop participants memorising an idealised answer. Depending, on the participants' preference
quizzes were completed either manually or on the computer. The quizzes were undertaken independently and took approximately 20 minutes to complete.

In addition, video recordings were taken for each session to allow for a more qualitative data collection. The videos were primarily taken to allow for analysis on levels of engagement and
motivation throughout the sessions. They also allow for other data collection such as discussions with other participants, talking out loud phases and other interactions with the sessions which could be missed when running a busy session.

For the final multiple choice quiz section of St. Vincent's Volcano, the participants could 'opt-in'
to having their progress recorded automatically through in-built game analytics. Gaming analytics can help identify the subject areas where participants are weakest, therefore allowing for a more tailored education session, as well as providing information about how the game is played and how it can be improved further. When 'opted in', information on how long each participant spent playing the game, the questions they were asked and how they answered
them was recorded. Unfortunately, for many of the student participants the analytics malfunctioned however, most of the adult participant's analytics were successfully recorded.

To have confidence in the evaluation of St Vincent's Volcano as a learning tool, the quality of the participant data is paramount. Although 126 school students were involved in the study, 31
participant datasets were removed from the study due to incompleteness of quizzes or lack of consent for use of the data and 22 datasets were removed due to evidence of cheating (using the internet) or copying from neighbours during the study. In addition, one of the school sessions with 8 students was used as a control experiment; these students only received the


outreach presentation and did not play the game. In total, 65 fully completed datasets of the participants who played the game could be used in this study, representing 51% of the original participant pool. For the adult sessions, 2 datasets were removed (due to one not completing quiz B and one participant was observed using the internet to complete the quiz) meaning that 23 datasets were used for this study, comprising 19 (83%) female and 4 (17%) male participants.

## 5.   RESULTS

This section presents preliminary results of implementation testing on St. Vincent. As this research is ongoing, not all aspects of the data collection have been analysed. Instead, this section presents the preliminary results of the primary data collection through knowledge quizzes.

### 5.1   Knowledge quizzes


Both student and adult participants completed two knowledge quizzes during their session, before and after the tests involving the St. Vincent's Volcano game and/or a presentation about volcanoes and St. Vincent. The percentage of completion for the two quizzes was measured with an increase in the percentage of questions answered between Quiz A and Quiz B by 12.8%
for student participants and 17.7% for adult participants.

All quizzes were coded using unique references for the participants to remove the chance of bias during marking. Marking of the quizzes was completed using an ideal answers template which allowed for the correct answers to be awarded points and further points to be awarded
where a deeper knowledge was demonstrated. The use of a template allowed for a uniform approach to the marking. The two scores for the pre-test and post-test quizzes were then plotted against each other as percentages for both the adult data sets and the student data sets. The results of the graphs are displayed in Figure 4.

*Figure 4. Graphs showing the pre-test and post-test quiz score marks in percentage for (a)Student participants and (b) Adult participants. The dashed line represents the line of improvement between the two tests above which there is knowledge improvement*

Both graphs show a general positive trend demonstrating that the participants' knowledge
improved after they had received an education session. The dashed line on both graphs is the line of improvement, in that any data points above this line indicate that they improved their knowledge between the two tests and any on or below this line did not improve their knowledge. For the adult data set there are 2 participants that fall on or below the line of improvement compared with 5 student learners. When calculated, the average percentage of
improvement for the adult participants is 9.3% compared to 8.3% for the student participants.

The $R^2$ value for the two graphs is also displayed on the graphs. For the adult set the $R^2$ is relatively low (0.39) showing a week correlation between the data. The $R^2$ value is higher for the student data set at 0.46 with this data set showing a stronger correlation.


To assess an individual's learning gain, we adopt Hake's (1998) normalised technique of calculating learner's knowledge change. Figure 5 shows the pre-test scores (%) for both adult and student participants plotted against their 'learning gain' - expressed as the difference between pre-test and post-test score % divided by the difference between the maximum


possible score (100%) and the pre-test score. This method determines potential 'gains' each participant can make irrespective of their initial starting level.

The average learning gains has also been calculated for the student data set and for the adult data set. Learning gains for the adult participants (0.11 ± 0.07) was slightly higher than the

average learning gains for the student participants (0.09 ± 0.06). These results demonstrate that the adult participants marginally benefitted more from the use of the game than the student participants.

*Figure 5. Graph showing the 'Learning Gains' for both adult and student participants and the*

*averages for both populations. Overall, both sets of participants show positive learning gains. The adult average is slightly more positive than the student average, indicating a greater improvement in learning gains.*

## 6.    DISCUSSION OF RESULTS


Initial results for the field testing of the St. Vincent's Volcano game demonstrate a general improvement for both adults and student participants. Adults had a marginally higher average improvement between the pre-test and post-test knowledge quizzes - 9.3% compared to 8.3% for the student participants. The same tendency is observed when the pre-test score percentage

is plotted against learning gains (Figure 5), indicating that the average learning gains for the adults is marginally higher than that of the students. This is a commonly observed trend for participants who have undertaken any form of outreach sessions (Ronan and Johnson, 2003, Shaw et al, 2004, Johnston et al, 1999; Paton et al, 2008). To establish the effectiveness of the game as an education and communication tool, it must be compared to an existing technique.


A comparison between a game session and a traditional session is only possible for the student participants. One session for student participants was undertaken comprising a presentation only (traditional method). From this session 8 student data sets were recorded, which demonstrate average learning gains of 0.12 ± 0.05, slightly higher than that of the students that

played the game (0.09 ± 0.06). The average percentage increase between the two quizzes was also higher for the traditional outreach student participants at 10.9%. Although this is an interesting finding, further research is required to understand the extent of effectiveness for the game as a stand-alone education tool compared to a traditional outreach session.


During the implementation, 5 (7.7%) student participants and 2 (8.7%) adult participants showed no improvement or even negative gains between the two quizzes. It is expected that this is a relic of the testing strategy rather than the game implementation. It was observed that some participants became despondent with the experiment due to its duration and the

repetitive nature of completing the knowledge quizzes. Video recording taken during the sessions will examine in more detail these contrasting levels of motivation and engagement and may be able to confirm this suspicion.

Of interest from the data collection is the number of participant's data that was removed due to

cheating or copying from a neighbour (particularly amongst the student participants). The number of data sets removed for this reason was 27.7% of the original dataset. Many of these



had used the internet to source answers. This could add weight to anecdotal arguments that a new generation of leaners exist in today's classrooms, where when an answer is unknown, the knowledge is instantly acquired from digital sources.


A general trend of improved knowledge is observed for 94% of the participants tested from both student and adult data sets. Further research is required to understand the full potential of using computer games in outreach which can be done using other data collected during the sessions (video recordings and game analytics). Present thinking, based on preliminary results,

suggests that the volcano game may be more effective when used in a contextual environment, supported with traditional outreach techniques such as a presentation. This will be further analysed as part of this ongoing study.

## 7. CONCLUSION


The preliminary findings from this study suggest that serious games have the potential to be effective tools in volcano education for both traditional stakeholder groups (school students) and also for non-traditional stakeholders (working-age adults). Although serious games would seem to be a highly promising communication and educational technique, this novel approach

faces a number of challenges. For one thing, game development is an expensive and time consuming process. The St. Vincent's Volcano game was the work of an in-house team of developers at Plymouth University over a one year period, during which it evolved via multiple iterations as the limitations of the software (and the designers) were overcome. Technical challenges ranged from the ability to correctly and realistically visualise the hazardous

phenomena to the interactivity and flow of the game. Although the completed volcano game was not intended to be of the equivalent quality as commercial computer games, the finished product still had to be fully functional, robust and interesting to play, and overall that requires a development process that is longer, more complicated, and more costly than conventional volcano awareness-raising measures.


Nevertheless, the early indications from this study are that volcano-based computer games hold much promise for hazard communication. Initial findings suggest they are most effective when used alongside conventional volcano outreach programmes, rather than being an alternative or a replacement activity. But while the more traditional methods of education, such as maps and

leaflets, typically target particular groups (especially school-aged children), the widespread popularity of computer gaming offers the opportunity to extend the reach to older and harder-to-reach demographic groups. This relates to the growing recognition that with advancements in technology, people today are taking in information in different ways to their predecessors. Perhaps more significantly, it is often difficult to establish the effectiveness of outreach sessions

or provide any follow-up to consolidate learning due to time and funding constraints of outreach organisations, but bespoke computer games - and their analytics within - open up the prospect of evaluating the effectiveness of hazard awareness and disaster preparedness.

The St. Vincent's Volcano Serious Game is an attempt to bridge the gap between these different

inter-generational learning styles and also to overcome some of the problems commonly encountered in conventional volcanic hazard education. Its twin aims are to improve knowledge of future potential volcanic hazards on St. Vincent and to integrate traditional methods of education in a more interactive manner. Although designed to support a pre-existing outreach volcanic hazard programme, our data suggests that the game could also be effective in




improving knowledge about volcanic hazards as a stand-alone tool. The ongoing research in
        this study will further refine the application of serious games to volcanic hazard communication,
        but nevertheless we feel confident that the virtual and immersive worlds of geo-gaming offers
        exciting opportunities to build knowledge and resilience among a diverse range of social
        groups within at-risk communities.

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

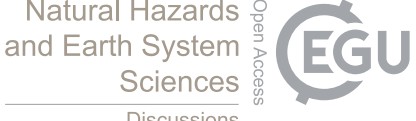



Table 1

| Topic | Stakeholder Answer | Community Answer |
|---|---|---|
| *Duration* | 15-30 minutes | 1 hour or more |
| *Platform* | Stand-alone application on mobile devices and laptop/PC | All available platforms (PC/laptop, mobile devices, internet, social media). |
| *Target audience* | Younger community members | Primary and secondary school children |
| *Content* | Volcanic phenomena (ash fall, pyroclastic flows and lahars), historical eruptions. | Historical eruptions and community response. |
| *Other* | Game should be used in current outreach sessions. | Game should be Free. |


Figure 1

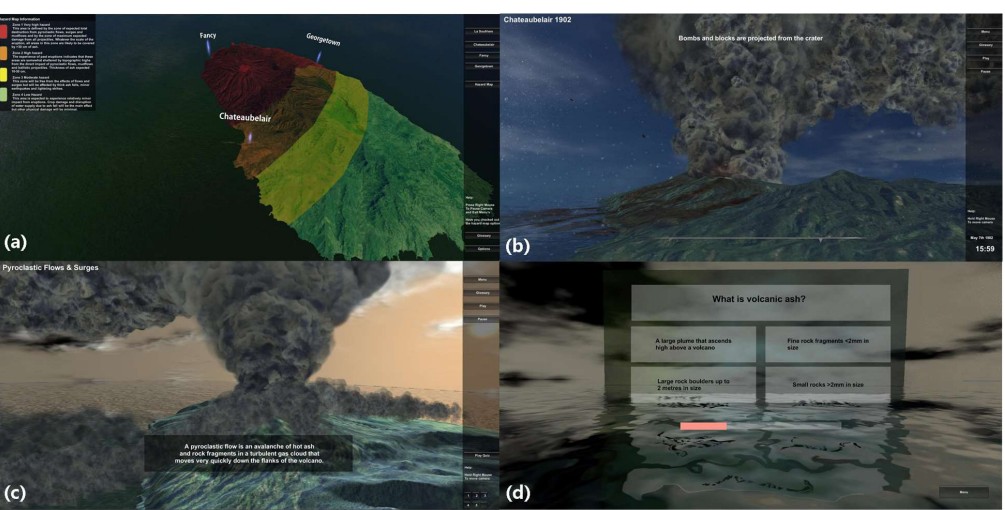





Figure 2

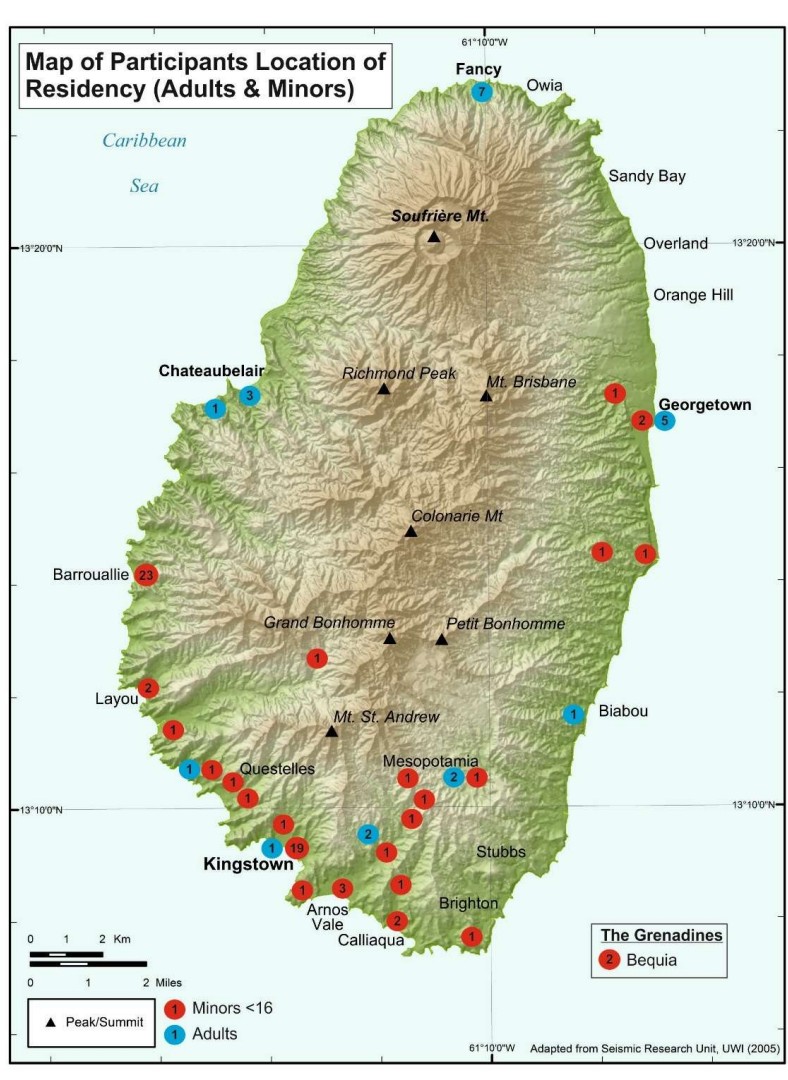




Figure 3

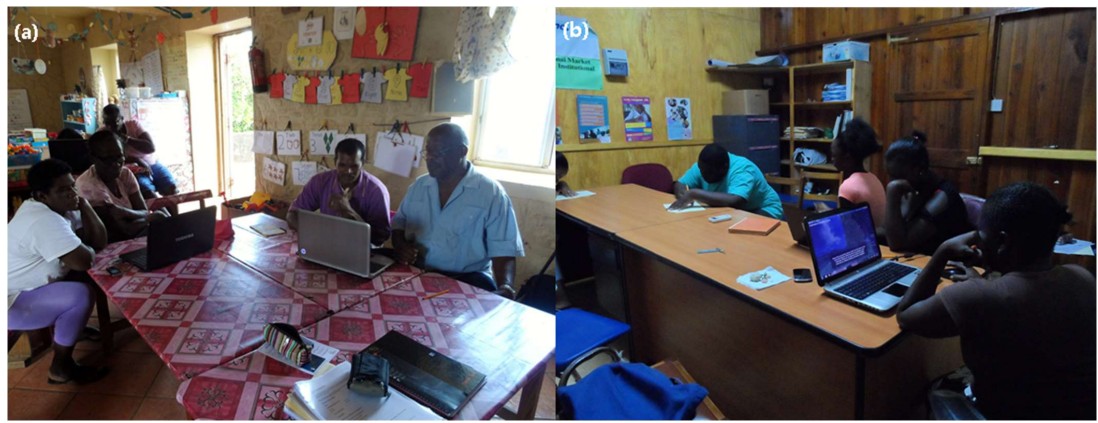



Figure 4

![Figure 4: Scatter plots showing (a) Students pre-test vs. post-test quiz scores with R² = 0.4651, and (b) Adult pre-test vs. post-test quiz scores with R² = 0.3934. Both plots show Post-test Quiz Score (%) on the y-axis and Pre-test Quiz Score (%) on the x-axis.]






Figure 5

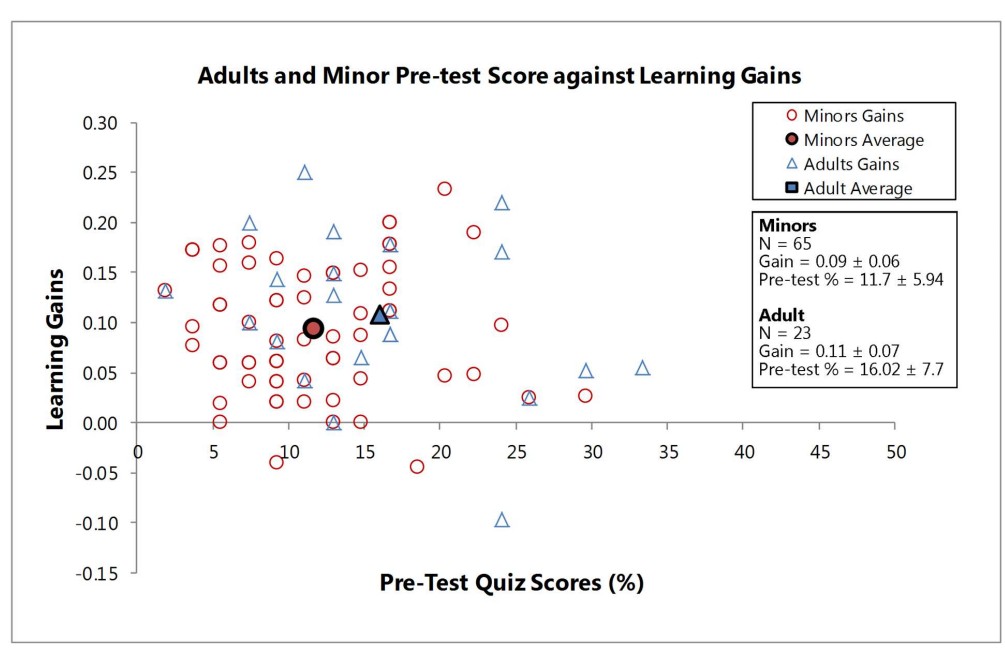