# Peer review of "Using video games for volcanic hazard education and communication."

_Natural Hazards and Earth System Sciences, 2016_

## Referee Comment (RC1) · S. Anderson (Referee) · 9 Feb 2016

The topic at the heart of the research paper is an important one - how can we effectively educate the public with respect to geologic hazards. The approach presented in this paper - using video games as a mechanism for better understanding - is well-grounded and justified. However, the methods used in this study (to infer that understanding for students and adults was enhanced through the use of the video game) may not be valid and reliable and therefore it is impossible to know if the measurement of learning reported here is truly learning (and not just an increase because of a flawed assessment instrument) and repeatable.

The authors state correctly that the "main source of data collection was through the completion of knowledge quizzes before and after all of the sessions took place". How-

ever, the actual assessment instrument was not contained in the manuscript so it was not possible to assess the main instrument of data collection. I infer from statements later in the manuscript that the instrument was multiple choice in nature, and there are well-known issues with using multiple-choice instruments for measuring learning. Julie Libarkin and her colleagues have published numerous studies over the past 15 years discussing the creation of multiple choice learning assessments for the geosciences that are both valid and reliable, and I would urge the authors to use these works as a guide for validating and modifying their test before they publish the data. In short, we can't be sure that the small gains shown on the exams are due to learning - the increases could be due to a number of issues related to validity and reliability discussed in the Libarkin papers.

In addition, it will probably be necessary to use the qualitative data that were collected, but not presented or discussed, to help ensure that that exams are valid and reliable. Pairing the qualitative with the quantitative data is an approach commonly used to create valid and reliable concept inventories in most scientific disciplines.

I found the discussion a bit underwhelming. Although the links to other studies were informative (for example, relating the greater gains for adult learners to other studies that found a similar result) there wasn't much discussion of how the students and adult learners were affected by the use of the game (other than just the increased post-test scores). I believe that the qualitative data that were collected but not presented should shed some light on some other changes that could occur through using the serious game (such as in increased or decreased desire to learn more about a volcano that has erupted or not erupted during their lifetime). In other words, is knowledge the only thing that changes as a result of the exercise, or are their some affective changes that also occur?

In summary, the topic of the paper, and the general approach (test an educational intervention with pre- and post- knowledge tests) is acceptable, but the assessment instrument has to go through a host of validity and reliability checks before we can

have any confidence that any change evident in the data truly represents learning. Libarkin outlines a number of potential approaches that the authors can use to create a research-quality assessment exam. I also urge them to include the qualitative data that should help to not only validate their knowledge test, but also shed light on any learning that may occur by using the serious game. I realize that this will delay the publication of the work here substantially, but in the end it is far better to publish a study that truly tackles the issue in a way that is valid and repeatable than it is to simply have another seldom-read education paper in the world.

---

## Author Comment (AC1) · 17 Feb 2016

Thank you for your comments about this manuscript. We suggest the following in response to comments made.

All the 'knowledge quizzes' used in this study comprised open ended questions, not multiple-choice as assumed by the reviewer. The multiple choice questions referred to in the manuscript relate to the interactive quiz contained within the game itself which is based on information provided throughout the game, and used as a method to reinforce the key themes rather than as a method of assessment.

The knowledge quizzes used for the pre and post testing comprised 12 open-ended questions which were developed by the author, and revised and refined in collaboration with a social scientist and a physical volcanologist. The questions were designed

to establish the basic level of knowledge of the participants but also to determine if there are common misconceptions within their knowledge and to any identify colloquial language and terminologies that may be used.

We realise the nature of the quizzes were not made clear in the methodology section and we will clarify this issue during revision. To avoid further confusion we will also use the terms 'multiple-choice volcano quiz' for the in-game quiz and 'knowledge test' for the pre and post-test assessments of which the data has been used.

As suggested by the reviewer the pre/post-session knowledge test will be included in the manuscript as an appendix for reference (see supplement).

We take on board the comments made by the reviewer in respect to the inclusion of qualitative data to add further validity to the study, and highlight any learning that may have occurred.

The primary aim of this paper is to assess the technique of using video games for this specific application and to present supporting data, as opposed to an exhaustive study. In response to this point we will include further qualitative data from the studies (e.g. video analysis of sessions) to strengthen the arguments presented.

Please also note the supplement to this comment:
http://www.nat-hazards-earth-syst-sci-discuss.net/nhess-2016-23/nhess-2016-23-AC1-supplement.pdf

**Supplement:**

**Volcano Pre-Test**
***Students***

[Figure]

The following quiz contains 12 questions about volcanic hazards. Please complete *all questions* with as much detail as you can. You can also use diagrams to answer the questions.

If you require any assistance, please ask a member of the outreach team.
* * *
**Questions**

1. What is the name of the volcano on St. Vincent?

2. What year did this volcano last erupt?

3. Can you name any other years when this volcano has erupted?

4. Volcanoes have many 'volcanic hazards' that occur during eruptions. What does the phrase 'volcanic hazard' mean?

5. Volcanic ash is produced by many eruptions around the world including those on St. Vincent but what exactly is volcanic ash?

6. Where does volcanic ash come from?

7. Lahars are common during and after volcanic eruptions. Why do lahars form?

8. How long after a volcanic eruption can lahars occur?

9. What is a pyroclastic flow?

10. Why are pyroclastic flows dangerous?

11. What colour zone of the 'volcanic hazard map' are these towns in? (Red, Orange, Yellow or Green).

    Chateaubelair

    Fancy

    Georgetown

12. How long can a volcanic eruption last?

---

## Referee Comment (RC2) · Solmaz Mohadjer (Referee) · 18 Feb 2016

This paper describes a pilot study that aims to examine the effectiveness of a computer game (St. Vincent's Volcano) in communicating volcanic hazards to residents of St. Vincent Island. Several studies have explored the use of video games as a learning approach in the context of geohazards. However, as pointed out by authors of this paper, there is often a lack of empirical evidence concerning the effectiveness of this method for teaching and learning about geohazards. The authors provide a good overview of the game design and the background knowledge on virtual reality environment and its potential in disaster risk reduction. The assessment approach, however, is not rigorous enough to evaluate the effectiveness of the video game as a learning technique. The only data discussed in this study are from pre- and post-quizzes. It is not clear what questions were asked and what objective(s) each question seeks to ad-

dress. Without this knowledge, it is difficult to state what the learning gain represents (for instance, rote memorization or deep understanding?). I suggest to the authors to make all quiz questions available, describe their objectives, and discuss results in relation to each question and what it may represent. I also recommend inclusion of data from video recordings as this often captures information that cannot be collected using quizzes. Data collected via in-built analytics could help answer questions concerning participants' performance and technical difficulties they might have encountered. Taken together, this paper touches on an important topic but without a systematic and rigorous assessment method, it would be very difficult to make a meaningful conclusion about participants' learning and the effectiveness of this method in communicating hazard-related information.

Here are some recommendations:

Page 1 line 37- Delete the extra semicolon.

Page 1 line 38- Define and spell out N-Gen or Net-Gen. Not everyone is familiar with these terms.

Page 2 line 71- Be consistent with the usage of 2-D/3-D. Sometime it is shown as 2D/3D in this paper.

Page 2 line 89- Insert a comma in the reference.

Page 2 line 95- Delete 'of' in '. . .we can use of virtual reality. . .'

Page 3 line 98- Place the period before the quotation mark (. . .learning.").

Page 4 line 153- Be consistent with how you spell St. Vincent. Sometimes it is shown with no dot after 'St' in this paper.

Page 4 line 168- What kind of instruments? Be more specific.

Page 5 line 201- It says later sections identify technical improvements for future iterations of the game. These technical improvements, however, are not stated clearly in the

paper. What kinds of data were collected to highlight technical difficulties? In Page 8, you mention the built-in analytics malfunctioned. Perhaps, this is an example of a technical issue that can be discussed? In Page 7, you mention students were able to ask questions throughout the session. Perhaps, some of their questions highlight technical issues associated with the game? Were these questions collected and processed?

Page 5 line 210- It would be really useful to know what questions were included in the online questionnaire. Perhaps this can be added as an appendix or supplementary information?

Page 5 lines 209-210- Please state who the end users are.

Page 5 lines 217-227- This paragraph describes the concept of the game based on the information received from focus groups and stakeholders. There is no mention of 'community response' in the concept despite being identified as something that should be covered in the content by the focus groups (see Table 1, Page 15). Please explain if and how this component (i.e., community response) was added to the content of the game.

Page 5 lines 229-235- Is it possible to have an image or a flow diagram that explains this paragraph in a visual way? It is a bit hard to follow.

Page 6 lines 242-243- The game follows Kolb's model with 4 stages of learning cycle that are mentioned on line 242. If this study uses this model, it would be useful to clearly connect each learning cycle (e.g., reflective observation) with the game design and implementation.

Page 6 – General question and a comment: Is this game open access and available online? It would be incredibly useful if it can be accessed online free of charge. Users can be asked to complete a feedback form to share their experiences and help with improving the game.

Page 6 line 269- Specify the type of elevation data and satellite imagery used in this

game.

Page 6 line 271- The game scene is said to be highly interactive. I suggest deleting 'highly' since it appears that it only allows users to add a hazard map and identify their location.

Page 6 lines 279-281- An example and/or image would be useful when describing the five clickable icons in each scene and the information each reveals.

Page 6 lines 282-283- What kind of questions were asked in the multiple-choice quiz? How were these questions designed? In other words, what are the objectives behind each question? It would be useful to include this information as supplementary information.

Page 7 lines 302-303- Change to "(based on the volcanic hazard map of Robertson, 2005). Delete the extra parenthesis.

Page 7 section 4.1- Mention the number of students who participated in this study (68 students?).

Page 7 line 302- Participants are mostly from schools located in the 'green hazard zone'. Does this affect your data and how?

Page 7 section 4.2- What is the age and background of the adult learners? Was this information collected? It would be useful to describe your adult learner population since this may play a role in explaining your results.

Page 7 line 328- It says 25 participants, but map shows 23 participants.

Page 8 Figure 3- I suggest deleting these photos as they do not add much to the paper. If you decide to keep them, I suggest showing one photo of the student session and one photo of the adult session, if possible.

Page 8 line 339- It says 'assistance was given to those that found difficulty with using the game'. It would be very interesting to know what kind of difficulty participants

experienced. This could help with identifying technical or other types of issues that can be fixed in the next implementation of the game. Were such data collected? If so, please consider including them.

Page 8 section 4.3- It might be useful to know the amount of time between the end of each session and when the post-quiz was administered. Was the post-quiz administered immediately after the session was over or a week later for instance? This might affect the results.

Page 8 section 4.3- Please consider discussing the data collected by video recording.

Page 8 section 4.3- Please explain the data recorded by in-built game analytics for the adult participants since the paper states that these data were collected successfully.

Page 9 line 382- Data sets for 19 female and 4 male student participants were used for analysis. Please report this number for the adult participants too.

Page 9 line 394- What are Quiz A and Quiz B? Are you referring to pre- and post-tests? If so, please use consistent terminology.

Page 9 line 400- It would be useful to know what questions require or reflect deeper knowledge. An example would be useful.

Page 9 lines 417-419- Report the significance for each correlation value and what it may indicate.

Page 10 line 437- Consider changing '…indicating a greater improvement in learning gains' to '…indicating a slight improvement in learning gains" since average learning gains are only slightly higher when comparing different data sets.

Page 10 section 6- Did the second presentation mentioned in the second paragraph contain identical content as the presentation used in the piloted educational session? If no, how did it differ and how does this difference affect the results?

Page 10 lines 462-463- There is no evidence in this paper to suggest that this statement

is true, especially when data from video recording are not included. I recommend deleting this statement if you do not plan to explain it with relevant data. That being said, it would be interesting to find out why these participants showed no improvement or even negative gains. Is it possible to state in what area(s) of knowledge (i.e. what questions) they showed no improvement? For instance, did they consistently provide wrong answers for questions of similar knowledge level?

Page 11 line 486- Change this sentence to "...suggest that serious games may have the potential..."

Page 11 line 502- Change this sentence to "...computer games might hold some promise for hazard..."

Page 11 line 515- In "...overcome some of the problems commonly encountered in conventional...," I suggest adding a few examples of these common problems.

Page 11 line 519- Change sentence to "our data suggest that the game could..."

Page 16 Figure 2- Consider adding an inset map to this figure to help readers with island location. I also suggest removing data not relevant to this study (e.g., peak names and names of towns not mentioned in the text). Label the volcano mentioned in the text by using the same name as the one used in the text (i.e., La Soufriere).

---

## Referee Comment (RC3) · Anonymous Referee #3 · 24 Feb 2016

Initial paragraph: This paper describes the steps to development of a computer-based game intended to "improve knowledge of future potential volcanic hazards on St. Vincent, and to integrate traditional methods of education in a more interactive manner." The work described is worthy; in the article it simply needs to be described as a work in progress so as not to mislead the reader. Additionally, methods for evaluation could be more thoroughly stated, and the authors should indicate how future game updates and implementation will be made and how the game will be integrated and sustained long-term in the education community. One systemic issue exists: the measuring of increased awareness (as opposed to possibly more easily measured 'preparedness') is problematic because it offers a lack of easily measurable variables. All that said, this is a commendable effort and descriptions of it in an article are worthy of publication with major modifications. This study adds to the long list of well-intended ventures that

face similar challenges.

Overall evaluation: The idea of using video games is a reasonable, if not trending approach to hazards education. As authors note in the Introduction, for the global hazards-education community, there is insufficient quantitative information available about the value of video games relative to other outreach methodologies. This study and paper have the potential to provide just that. The study and this paper serve as a caution for others considering similar ventures. Owing to a variety of unanticipated obstacles encountered in the video game development and implementation processes, this paper falls short in providing meaningful results. While the word 'preliminary' is used near the end of the Abstract, the reader must read most of the paper before truly recognizing that video game development and implementation are works in progress.

This reviewer hopes that the malfunctioning of analytic software, necessary tossing of much data owing to systemic student cheating, and other challenges encountered are not roadblocks, but valuable information for consideration during game upgrades and implementation during the future Volcano Awareness Weeks. Raising awareness and levels of preparedness are long-term propositions. This is simply round one. In that sense, the paper should be revised to (1) reflect from the start that this is a report of a work in progress; and (2) couch the statement of problems encountered as a lessons learned, and with recommendations for how they will fix these issues for the next round. Readers who are contemplating similar ventures will welcome a paragraph about 'lessons learned', and recommendations for the development and implementation processes. A change in title that reflects the 'in progress' and 'lessons learned' aspects would provide potential readers with a more accurate description of content; it would enhance and not detract from the article's value.

Readers would also welcome a broader description of the role of the video game in VAW, how they are creating long-term buy-in by educators, and plans for sustaining and upgrading the video game product over time.

[Figure]

Scientific comments (specific) In the Introduction and Previous Research sections, the authors include a helpful overview of the potential value of educational tools in the raising of awareness about hazards, and the need for products which are targeted for specific audiences. It explains well the value of targeting students in the larger quest to reach their families. There is an omission here that must be noted—that some of the same authors—Johnston and Paton in particular, note also in their professional publications that awareness does not equal preparedness, and that there is a cultural filter, and a series of mental steps through which information must progress before tangible results—preparedness—are achieved. This concept needs to be acknowledged more specifically within the text.

The experiment design makes full analysis unachievable—quantitatively—and in some ways qualitatively. The study objectives are noted, but defining and then working towards a tangible 'desired outcome' would have made success easier to measure. For example, authors could arrange for verification of the family creating a family emergency communication card, adding prescribed general items to an emergency kit, and adding an evacuation map/instructions to this kit.

Regardless of the lack of sufficient quantitative and qualitative assessments presented in this paper, there is still hope that the same verification processes can be accomplished for each future VAW event. Perhaps adjustments can be made by creating some tangible measures. The user can return to the same group year after year during Volcano Awareness Week and with an improved quantitative tool, and can still make those assessments. After all, raising awareness is broadly acknowledged as a long-term proposition, with one year serving as a first point in the data series.

Comments specific with a listing (by page and line) of technical questions. Page 1 Line 23—This is very preliminary data. Line 18 speaks of 'final implementation'; 23 calls it preliminary data. Throughout the Results there is reference to this being a work in progress. That should be noted in the abstract. Page 1 Line 37-remove extra semicolon. Page 3 Line 100—For others considering similar ventures, there would

be value in a brief description of the characteristics common of successful games. What are they? Which of the common characteristics that are shared with other products aimed at youth and adults, such as hazards-related educational comic books (also enjoying some popularity), creative play such as play-acting theatrical productions about hazards-preparedness-positive outcomes, school projects, etc. Page 5 Line 201—Readers need additional information about technical improvements to be made. Page 5 Line 210—It is impossible for the reader to analyze results without having the questions included. Page 7 Lines 300-325—It would be useful for the reader to learn more about the surrounding educational environment. For example, beyond the VAW outreach team efforts, did students receive any other volcano-related training in their school? See also comments in Overall Evaluation. Page 9 Line 380—Say up front that this is preliminary only. See also comments in Overall Evaluation. Page 10 Line 471—The high percentage of data sets removed serves as an indicator of where techniques for implementation should be modified in future efforts. This is valuable data; implementation during one VAW is simply is the first step. State how you will upgrade the game's analytics and implementation procedures in the future. Give guidance to readers who might attempt similar ventures. Page 11 Line 503—This statement about "Initial findings. . .." seems expectable, but I am searching for the analysis on which it is based. Please provide more information. Page 11 Line 518—There would be value in placing these aims much earlier in the paper. Figure 5—Correct the Explanation Box so that it shows the Adult Average as a blue triangle rather than square.
* * *

---

## Author Comment (AC2) · 14 Mar 2016

In the opening paragraph the reviewer suggests that the testing strategy and data provided within this paper is not sufficient enough to make meaningful conclusions about participants' learning and the effectiveness of this method in communicating hazard-related information. The authors would like to comment here that this study is currently a work-in-progress and is designed to provide an initial evaluation of the technique of using video games for education rather than as an exhaustive study. Further, this manuscript has been written to provide an insight into some of the key lessons learned from undertaking this kind of study for other researchers considering pursuing a similar line of research. We acknowledge that these aims may not be clear in the current manuscript but will clarify this in the finalised manuscript. Additionally, the title of the manuscript will be altered to reflect this issue.

We agree with the reviewer that the methodology currently described requires further detail and this will be added to the revised manuscript and a copy of the testing instrument will also be included. To clarify, the pre and post-test quizzes comprised 12 open-ended questions developed by the authors, and revised and refined after consultation with a social scientist.

With regards to the multiple-choice quiz, this is a section within the game itself and was not designed to be used to test for knowledge gain. Instead, data from this can identify areas where students are weakest and allows for educators to tailor outreach sessions to confront this.

To address the comments made about the technical difficulties encountered during the testing, the technical issues that were encountered, such as the failure of the game analytics, will be described in more detail in the revised manuscript.

Thank you for the detailed comments relating to minor changes required to the manuscript. These changes will be made to the final manuscript and further detail will be added where appropriate.

---

## Author Comment (AC3) · 14 Mar 2016

A stronger emphasis will be made throughout the finalised manuscript to clarify that this is a work-in-progress. Additionally, the title of the manuscript will be altered to reflect this and to also identify that this paper is an evaluation/overview of the technique.

Previous reviewers also suggested a more detailed methodology was needed and we acknowledge that the methodology needs to be clearer and with the knowledge quizzes included for reference.

A section will be added to the revised manuscript to discuss 'lessons learned' and for recommendations for other researchers pursuing a similar testing method.

In reference to the comments about Johnston and Paton's research and the realisation

that awareness does not always lead to preparedness: This section will be recast to acknowledge this concept within the manuscript.

In this instance, the game is designed to improve knowledge, language and understanding of volcanic phenomena for participants and not to be a comprehensive disaster risk reduction tool. For this reason, the method of measuring effectiveness through the adoption of preparative measures is not applicable as this information about this is not provided within the game itself. Instead, we have used a tried and tested method of testing to establish 'learning gains'. However, we realise that this message can be made clearer in the manuscript and a discussion of the reasons for the testing strategy utilised will add further clarity to this.

We thank you for your further comments in the specific section and will integrate these throughout the revised manuscript.